# Evaluating Different Extraction Approaches for GC-MS Based Metabolomics Analysis of the Giant Pandas’ Fur

**DOI:** 10.3390/toxics10110688

**Published:** 2022-11-14

**Authors:** Yang Yang, Yanqiang Yin, Xianglan Tang, Yinyin Xia, Jinya Zhang, Chun Yan, Weixuan Zhang, Hua Zhang, Ting-Li Han

**Affiliations:** 1Department of Obstetrics and Gynecology, The First Affiliated Hospital of Chongqing Medical University, Chongqing 400016, China; 2Ministry of Education of China International Collaborative Joint Laboratory of Reproduction and Development, Chongqing Medical University, Chongqing 400000, China; 3State Key Laboratory of Maternal and Fetal Medicine of Chongqing Municipality, Chongqing Medical University, Chongqing 400000, China; 4Institute of Life Sciences, Chongqing Medical University, Chongqing 400000, China; 5Chongqing Zoo, Chongqing 400050, China; 6Department of Obstetrics and Gynecology, The Second Affiliated Hospital of Chongqing Medical University, Chongqing 400010, China; 7School of Public Health and Management, Chongqing Medical University, Chongqing 400000, China; 8School of Clinical Medicine, Chongqing Medical University, Chongqing 400000, China

**Keywords:** GC-MS, endogenous and exogenous metabolome, extraction method, giant pandas

## Abstract

Giant pandas in zoo captivity are situated in residential areas, where environmental pollutants and anthropogenic factors have an impact on their health. Hair metabolomics has been applied in numerous environmental toxicological studies. Therefore, the panda fur metabolome could be a reliable approach to reflect endogenous and exogenous metabolic changes related to environmental exposure. However, there is no established extraction protocol to study the fur metabolome of pandas. The aim of this research was to optimize the extraction of panda fur metabolome for high-throughput metabolomics analysis using gas chromatography-mass spectrometry. Fur samples were collected from five pandas. Eight different extraction methods were investigated and evaluated for their reproducibility, metabolite coverage, and extraction efficiency, particularly in relation to the biochemical compound classes such as amino acids, tricarboxylic acid cycle derivatives, fatty acids, and secondary metabolites. Our results demonstrated that HCl + ACN were the superior extraction solvents for amino acid and secondary metabolite extraction, and NaOH + MeOH was ideal for fatty acid extraction. Interestingly, the metabolomic analysis of panda fur was capable of discriminating the longitudinal metabolite profile between black and white furs. These extraction protocols can be used in future study protocols for the analysis of the fur metabolome in pandas.

## 1. Introduction

The giant panda (*Ailuropoda melanoleuca*) is an iconic flagship species and one of the world’s endangered animals. In order to protect the species, conservation zones and captive breeding centers have been established. In China approximately 422 captive pandas have been living in 68 zoos. However, recent studies have shown that conservation efforts are being compromised by increasing levels of pollution associated with China’s rapid industrialization and urbanization [1,2]. Therefore, the captive pandas living in zoos are close to human living environments and affected by anthropogenic factors and are exposed to higher concentrations of environmental toxins than are wild pandas. Pandas in Sichuan have been reported to be exposed to high levels of heavy metals associated with coal combustion, industry, and traffic [3]. Organic pollutants such as polychlorinated dibenzo-p-dioxins and dibenzofurans, polychlorinated biphenyls, and organochlorine pesticides were higher in samples from pandas held in captivity than from those living in nature reserves [1]. Chen et al. showed that environmental toxins compromised the proportion of live sperm, and the aberrance ratio of sperm was significantly higher for captive pandas than for wild pandas [1]. These studies demonstrated that environmental pollution has a great impact on the health of giant pandas. However, it is impractical to acquire fresh biopsies from large animals. Alternative non-invasive specimens are required to reflect the longitudinal exposure and physiological metabolism changes in pandas.

The regulatory mechanisms by which environmental toxicants influence cellular metabolism are an active field of research pertaining to environmental toxicology. Metabolomics has been increasingly applied in environmental toxicological studies, as it facilitates the screening of biological targets of environmental contaminants and yields understanding on different molecular levels related to endogenous processes and metabolism (biological perturbations) and external contributors to the exposome [4,5]. Together with modern mass spectrometry technology to accurately identify hundreds to thousands of small compounds in complex biofluids and tissue biopsies, metabolomics is the preferred analytical approach to study how panda metabolic change occurs in response to environmental perturbations. Moreover, the metabolomic analysis of pandas has been performed using conventional biological samples, including plasma, urine, and feces, but the fur metabolism study was still missing. Fur could be used as a valuable analytical sample to investigate retrospective xenobiotic exposure, as it provides for better detection of long-term exposure compared to other biological samples. Fur also has many advantages, including effortless sample collection, convenient transport, and storage at room temperature [6]. Moreover, fur also contains functional metabolomes such as amino acids and lipids that could reflect endogenous metabolism [7]. Similarly to the analysis of hair metabolome, it has been suggested that nails can be utilized as alternative specimens to evaluate long-term environmental exposures [8,9]. In addition, segmental analysis of hair based on its growth rate can provide information on metabolic changes over time. As for pandas, they generally molt their fur biannually around May and October in China. Therefore, panda fur has great potential as a metabolomics sample to monitor endo- and exogenous exposure longitudinally.

There have been many sample preparation methods reported for the metabolite profile of human hair [8]. However, how to identify metabolic analytes in panda fur remains unreported. There is a challenge in developing these methods derived from individual metabolites, as these metabolites exhibit their own unique chemical and physical properties (e.g., molecular weight, polarity, molecular size, volatility, stability, solubility, and many more) in the hair matrix. Therefore, we adopted several common methods of human hair sample preparation to study the metabolism of panda fur. Sodium hydroxide (NaOH) [9], and hydrochloric acid (HCl) [10,11] at high temperature were applied to digest the fur. These, together with rapid liquid/liquid extraction solvents, such as methanol [12,13], acetonitrile [14], or n-hexane/ethyl acetate [9], were used to extract the organic components. Furthermore, gas chromatography-mass spectrometry (GC-MS) based metabolomics is the most prevalent analytical platform to study human hair profiles [7,10,15,16,17,18]. This preference is likely due to its high separation capability in analyzing polar metabolites, even though these polar compounds must be derivatized prior to analysis [19]. Electron impact (EI) ionization in GC-MS generates highly reproducible mass spectra independent of vendors [19], making GC-MS libraries available for public sharing and exchange. Thus, the aim of the present work was to develop and validate an accurate sample preparation method for fur metabolism analysis to measure the exposome and metabolome of panda fur using GC-MS based on methyl chloroformate (MCF) derivatization. Our final objectives were (i) to find a suitable method with high accuracy, sensitivity, and specificity for the metabolism study of panda fur, and (ii), to establish a comprehensive protocol to study panda hair from sample collection, sample preparation, and chemical extraction.

## 2. Materials and Methods

### 2.1. Chemicals and Reagents

Sodium hydroxide (NaOH), potassium hydroxide (KOH), sodium bicarbonate (NaCO_3,_ 50 mM), anhydrous sodium sulfate, and 2,3,3,3-d4-alanine (D4-alanine, internal standard) were purchased from Sigma-Aldrich (St. Louis, USA). Hexane, ethyl acetate (EtOAc), and methanol (MeOH) were bought from Adamas-beta (Shanghai, China); acetonitrile from ThermoFisher (USA); hydrochloric acid (HCl) and chloroform from Chongqing Xinan Chemical Reagent Company (Chongqing, China); pyridine from Merck (Darmstadt, Germany); and methyl chloroformate (MCF) from Shandong Huayang Pesticide Chemical Industry Group (Shandong, China).

### 2.2. Sample Collection

The overall experimental design is summarized in Figure 1. The white and black fur samples were collected by certified zoo staff from the hindneck of five giant pandas raised at Chongqing Zoo, China. The process was approved by the Chongqing Zoo Ethics Committee. The furs were stored in aluminum foil at −20 °C.

### 2.3. Hair Segmentation and Decontamination Procedures

All fur samples were segmented by scissors and washed with 2 mL ultrapure water and 2 mL methanol twice. Five milligrams of fur were weighed and transferred into silanized glass tubes with a Teflon screw cap. To further validate our metabolomic methods, white and black panda fur samples were cut into three segments: the first segment (0–1 cm, 1st); the second segment (1–2 cm, 2nd); and the third segment (2–3 cm, 3rd).

### 2.4. Acid Hydrolysis

The segmented fur samples were acid-hydrolyzed by adding 400 μL of 6 M HCl [13] with 10 µL internal standard (L-alanine-2,3,3,3-d4 10 mM) [13] and incubated at 110 °C for 6 h (Heratherm OMA 100, Thermo, Germany). After the solutions were cooled to room temperature, they were filtered using 0.45 μm PVDF filters to remove precipitated lysates.

### 2.5. Base Hydrolysis

The segmented fur samples were base-hydrolyzed by adding 400 μL 1 M NaOH with 10 µL internal stand (L-alanine-2,3,3,3-d4 10 mM)) at 85 °C for 25 min [20] and cooled to room temperature. After the solutions were cooled to room temperature, they were filtered using 0.45 μm PVDF filters to remove precipitated lysates.

### 2.6. Liquid/Liquid Extraction

After acid and base hydrolysis, digested samples were added into one of the following three solvent extraction solutions (ACN [21], Hexane-EtOAc (Hexane: EtOAc = 9:1) [11,20], or MeOH [15]). The compositions of eight extraction reagents were as follows: (1) HCl + ACN: HCl and ACN in 4:1 ratio. (2) HCl + Hexane-EtOAc1: HCl and Hexane-EtOAc in 1:1 ratio. (3) HCl + Hexane-EtOAc4: HCl and Hexane-EtOAc in 1:4 ratio. (4) HCl + MeOH: HCl and MeOH in 2:5 ratio. (5) NaOH + ACN: NaOH and ACN in 4:1 ratio; (6) NaOH + Hexane-EtOAc1: NaOH and Hexane-EtOAc in 1:1 ratio. (7) NaOH + Hexane-EtOAc4: NaOH and Hexane-EtOAc in 1:4 ratio. (8) NaOH + MeOH: NaOH and MeOH in 2:5 ratio. Samples were centrifuged at 3220× *g* for 15 min, and the solution of the analyzed layer was dried by SpeedVac for 3 h. The dried hair extracts were stored at −80 °C until methyl chloroformate derivatization.

### 2.7. MCF Derivatization

Dried fur samples were derivatized using an MCF method according to Smart et al. [22]. In brief, 200 μL 1 M NaOH was added to resuspend, followed by 167 μL MeOH and 34 μL pyridine. Twenty microliters of MCF were added to initiate derivatization by vigorous mixing for 30 s, and then another 20 µL MCF were added, followed by another 30 s of mixing. A 400 µL amount of chloroform and 400 µL of 50 mM sodium bicarbonate were added and mixed for 10 s to separate the derivatized metabolites. The lower chloroform layer was used for GC/MS analysis.

### 2.8. GC/MS Analysis

Derivatized samples were analyzed on an Agilent GC7890 system coupled with electron impact ionization (70 eV). In brief, chromatographic separation was achieved using a ZB-1701 gas capillary column (30 m × 250 μm id × 0.15 μm with 5 m guard column, Phenomenex), with a constant helium gas flow of 1 mL/min. The parameters of the GC settings and MS were in accordance with Han et al. [23]. The temperature of the inlet, the auxiliary, MS quadrupole, and MS source were 290 °C, 250 °C, 230 °C, and 150 °C, respectively. The mass range was set from 30 µm to 550 µm. Scan speed was set at 1.562 µ.s^−1^, and the solvent delay was set to 5.5 min.

### 2.9. Metabolite Identification, Data Extraction, and Statistical Analysis

AMDIS software and our in-house MCF mass spectral library and NIST library were applied for metabolite deconvolution and identification. The compounds were identified according to two criteria: >90% library match to our library spectra, and within a 0.5 min bin of the respective chromatographic retention time. The relative metabolite concentrations were extracted by our in-house MassOmics software with the peak height of the highest reference ion [24]. The metabolite concentrations were normalized by the abundance of the D4-alanine internal standard and the weight of the fur. Statistical analyses were performed using Microsoft Excel and R software. The coefficient of variance (CV) was used to check the repeatability of extraction methods. The illustration of two-dimensional projections of principal component analysis (PCA), orthogonal partial least squares discriminant analysis (OPLS-DA), dot plots, histogram, Venn diagrams, heatmaps, and flower plots were generated by the ggplot2, UpSetR, ComplexHeatmap, and plotrix R packages.

## 3. Results

This study was the first to evaluate the protocols of different extraction solvents to be applied in analyzing the exposome and metabolome of panda fur using GC-MS. After digesting hair with alkaline or acidic hydrolysis, we evaluated the extraction efficiency of four different extraction solvents by testing extraction reproducibility, metabolite coverage, and recovery efficiency. The detailed information is displayed in Appendix A.

### 3.1. Analytical Qualities and Reproducibility of the Eight Different Extraction Methods

The representative total ion chromatograms (TIC) of the fur extraction resulting from the eight different extraction approaches are illustrated in Figure 2. After chromatographic peak deconvolution and identification, 77 fur metabolites were identified using our in-house MS library of chemical standards (Appendix A). Nevertheless, glutamine and arginine levels in panda fur were below the detection limit of our method. HCl + Hexane-EtOAc 1, HCl + ACN, HCl + Hexane-EtOAc 4, and NaOH + MeOH showed the most promising extraction reproducibility for giant pandas, respectively (Figure 3). The PCA analysis demonstrated the technical replicates of HCl + Hexane-EtOAc 1 fur ex-tracts (light green dots), HCl + ACN fur extracts (red dots), HCl + Hexane-EtOAc 4 fur extracts (yellow dots), and NaOH + MeOH fur extracts (purple dots) clustered the closest to each other along PC1 (53.3%) and PC2 (26.1%) (Figure 3a). Consistent with the results of the PCA, HCl + ACN, HCl + Hexane-EtOAc 1, HCl + Hexane-EtOAc 4, and NaOH + MeOH exhibited the most reproducible accumulated coefficients of variance (CV: 61.5%, 66.0%, 67.2%, 69.5%) overall for the biochemical classes identified in panda fur, respectively (Figure 3b). Interestingly, the reproducibility of amino acid was exceptional among the different methods when applied with acid hydrolysis (number:18, CV:3.4–5.2%, Appendix A).

### 3.2. Metabolite Coverage of the Eight Different Extraction Methods for Panda Fur

The second criterion we used to determine the optimal extraction solvent was the metabolite coverage in the fur metabolome of giant pandas. The flower diagram shows the total number of identified metabolites in each extraction solvent for the giant panda fur samples (Figure 4b). These were NaOH + ACN (54 metabolites), HCl + ACN (53 metabolites), HCl + Hexane-EtOAc 1 (52 metabolites), NaOH + MeOH (52 metabolites), NaOH + Hexane-EtOAc 4 (48 metabolites), HCl + Hexane-EtOAc 4 (47 metabolites), HCl + MeOH (46 metabolites), and NaOH + Hexane-EtOAc 1 (46 metabolites) (Figure 4b). Furthermore, UpSet plots (Figure 4a) illustrated that 22 common metabolites were detected in eight different extraction methods, and 17 of these metabolites were amino acids, as exhibited in the A list (Figure 4a). Seven metabolites were detected by hydrolysis methods, as displayed in the G list (Figure 4a). Most of them were saturated fatty acids, including citramalic acid, heneicosanoic acid, margaric acid, myristic acid, and trans-vaccenic acid. Five metabolites were only detected by acid-hydrolyzed methods, as shown in the H list (Figure 4a), including homocysteine, 2-aminoadipic acid, 4-methyl-2-oxopentanoic acid, levulinic acid, and D-citraconic acid.

### 3.3. Extraction Efficiency of Different Extraction Methods

The last criterion for the evaluation of the extraction solvents was extraction efficiency, which was determined by the sum of all peak intensities of extracted metabolites in a similar biochemical class, as shown by Figure 5 and Appendix A. All in all, for the panda fur metabolome, acid hydrolysis displayed the highest peak intensity for amino acids, amino acids and derivatives, and secondary metabolites (most efficient extraction: HCl + ACN). Furthermore, base hydrolysis produced the effective extraction yield for saturated fatty acid (most efficient extraction: NaOH + MeOH). It has been reported that acidic ACN hydrolysis is effective in liberating amino acids from proteins [25]. Aqueous ACN helps to weaken the hydrophobic interactions and enhance the peptide–peptide hydrogen bond, leading to the denaturation of proteins [25]. Then, the peptide bonds (H–N–C=O) forming the polymer chain of hair keratin can be readily cleaved into a mixture of single amino acids by HCl [26]. Furthermore, base hydrolysis followed by methylation is a recommended method for routine fatty acid analysis. Ester bonds between the fatty acids and glycerol of a triglyceride are cleaved by a hydration reaction via free hydroxide released from the base. Subsequently, the hydroxyl group of fatty acid is replaced by a methoxy ion from methanol [27]. Thus, our suggested methods agree with the biochemical nature of amino acids and fatty acids.

### 3.4. Validation Experiment: Metabolite Profile of Three Segments between White and Black Panda Fur

In order to validate our suggested HCl-ACN method, the metabolic profiles of white and black panda fur in different segments using HCl-ACN extraction and GC-MS analysis were identified. PLSDA analysis illustrated that the white and black fur were separated, and the R^2^ and Q^2^ validations of the OPLS-DA model were 0.61 and 0.64 for two accumulated principal components, respectively (Figure 6a). Based on GC-MS analysis, a total of 30 metabolites were found to be significantly altered either when transitioning from first segment to second segment or second to third segment between black and white panda fur (Figure 6b). This demonstrated that the metabolic profile of the hair changed substantially between different colored fur and segments. The majority of significantly different metabolites between black and white fur were amino acid and derivatives, organic acids, saturated fatty acids, secondary metabolites, and TCA cycle and derivatives. In the black 1st segment, the levels of amino acids such as asparagine, glutamic acid, histidine, isoleucine, lysine, methionine, phenylalanine, serine, threonine, and tyrosine were higher, while only cysteine and tryptophan were decreased (Figure 6c). Borges et al. demonstrated that human black hair contains brown-black eumelanin, a polyanionic indolequinone-based polymer, which facilitates hair matrix attraction with other metabolites or drugs through ionic interactions [28]. Thus, black fur may exhibit a higher affinity for retention of metabolites than white fur. Therefore, the panda fur metabolite profile could possibly reflect the panda’s metabolism in response to long-term exposure and physiological changes. Hence, panda fur has great potential as a metabolomics sample to monitor endo- and exogenous exposures longitudinally.

## 4. Conclusions

In summary, this is the first study to investigate the optimal extraction solvent for analyzing the global fur metabolome of giant pandas by GC-MS analysis. Based on the criteria of reproducibility of extraction, detected metabolite coverage, and the extraction recovery yield, HCl + CAN was the superior extraction solvent for amino acid and secondary metabolite extraction, and NaOH + MeOH was ideal for fatty acid extraction. However, different organic solvents favored the extraction of specific biochemical classes. Future studies need to carefully select their extraction solvents to meet the primary purpose of hair metabolome study. The metabolic profiles of three segments between white and blank panda fur showed that panda fur has great potential as a metabolomics sample to monitor endo- and exogenous exposure longitudinally.

## Figures and Tables

**Figure 1 toxics-10-00688-f001:**
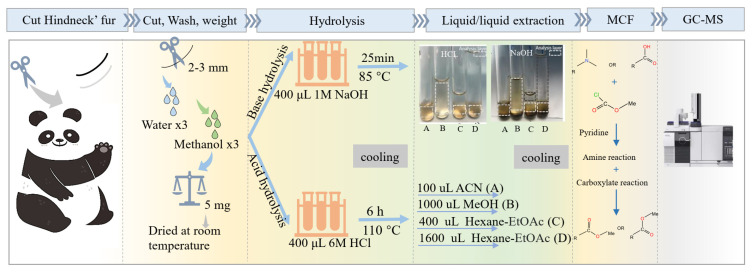
The overall workflow of method development for giant panda fur metabolomic study. The first step was to collect panda fur samples from the hindneck, and then the samples were cut, washed, weighed, and dried at room temperature. The dried fur samples underwent either basic or acidic hydrolysis, followed by four different organic solvent extractions, with subsequent drying by a SpeedVac. The dried extracts were derivatized using a methyl chloroformate (MCF) method, and the lower chloroform layer was used for GC/MS analysis. NaOH = sodium hydroxide, HCl = hydrochloric acid, ACN = acetonitrile, MeOH = methanol, EtOAc = ethyl acetate, MCF = methyl chloroformate, GC/MS = gas chromatography-tandem mass spectrometry.

**Figure 2 toxics-10-00688-f002:**
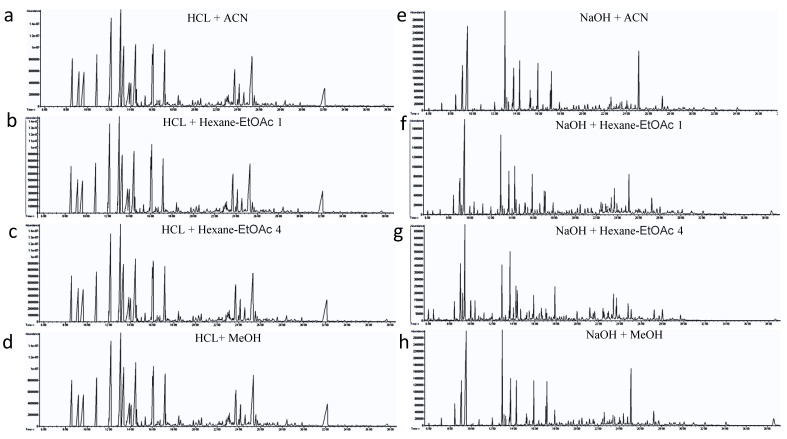
Representative total ion chromatograms (TICs) of the fur metabolome for giant pandas, using eight different extraction methods. (**a**): HCl: ACN in 4:1 ratio (HCl + ACN); (**b**): HCl: Hexane-EtOAc in 1:1 ratio (HCl + Hexane-EtOAc 1); (**c**): HCl: Hexane_EA in 1:4 ratio (HCl + Hexane-EtOAc 4); (**d**): HCl: MeOH in 2:5 ratio (HCl + MeOH); (**e**): NaOH: ACN in 4:1 ratio (NaOH + ACN); (**f**): NaOH: Hexane-EtOAc in 1:1 ratio (NaOH + Hexane-EtOAc 1); (**g**): NaOH: Hexane-EtOAc in 1:4 ratio (NaOH + Hexane-EtOAc 4); (**h**): NaOH: MeOH in 2:5 ratio (NaOH + MeOH).

**Figure 3 toxics-10-00688-f003:**
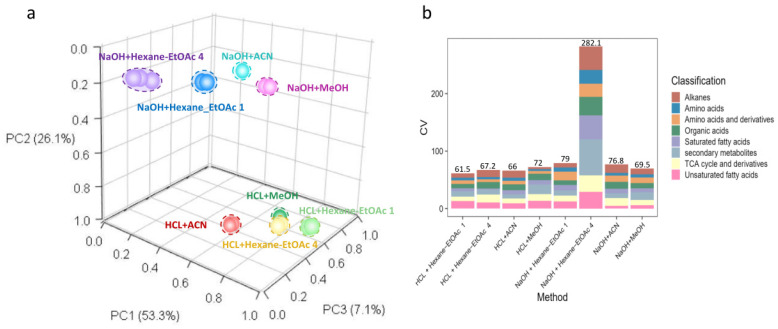
(**a**): Principal component analysis (PCA) of giant panda fur metabolomes extracted using eight different solvents. (**b**): The bar chart represents the cumulative coefficients of variation (CV) of eight different biochemical classes using eight different extraction solvents. The major metabolite classifications were alkanes, amino acids, amino acids and derivatives, organic acids, saturated fatty acids, unsaturated fatty acids, secondary metabolites, and TCA cycle derivatives. The *y*-axis is the total CV for all major metabolite classes.

**Figure 4 toxics-10-00688-f004:**
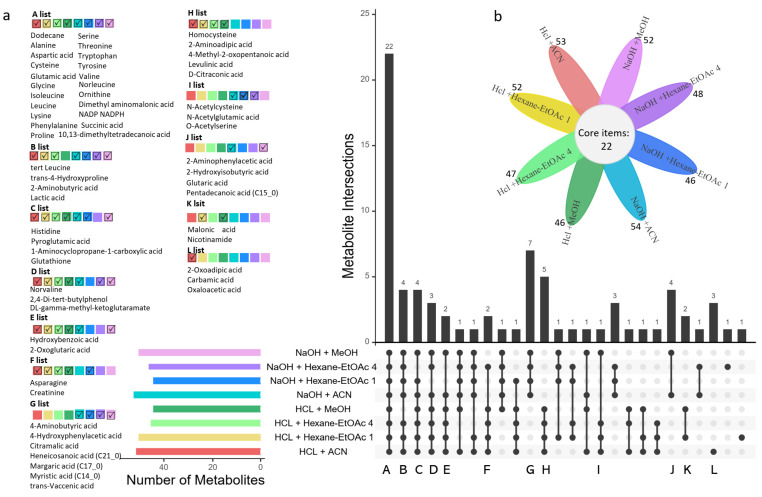
Metabolite coverage compared across eight different extraction methods in panda fur. (**a**) UpSet plot presents relationships between interactive sets of eight different extraction methods. The horizontal bar graph indicates the number of metabolites detected in each extraction method. The individual or connected black dots indicate the metabolite intersections that were either unique or shared between extraction methods. The vertical bar graph represents the number of shared metabolites. The metabolite list A–L displays shared or unique metabolites between extraction methods, as shown by checked boxes (√). (**b**) The flower plot shows the number of identified metabolites for each extraction method. Red: HCl + ACN; yellow: HCl + Hexane-EtOAc 1; light green: HCl + Hexane-EtOAc 4; dark green: HCl + MeOH; cyan: NaOH +ACN; blue: NaOH + Hexane-EtOAc 1; purple: NaOH + Hexane-EtOAc 4; and red violet: NaOH +MeOH.

**Figure 5 toxics-10-00688-f005:**
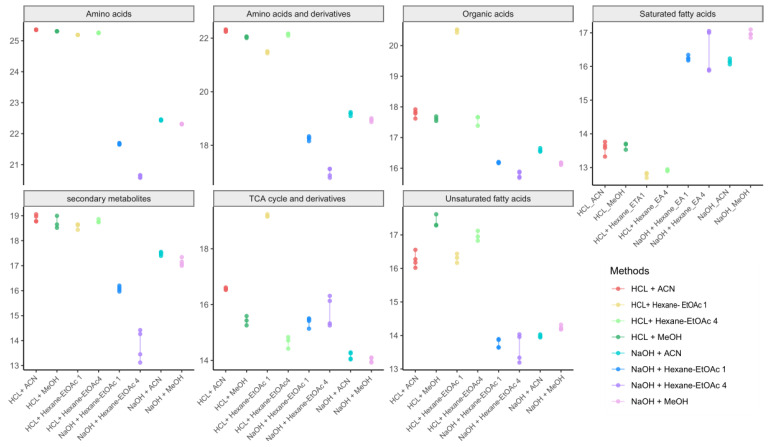
The dot-line graphs show the total concentrations of metabolites in each metabolic biochemical class across eight different extraction methods for giant panda fur samples. Each dot represents a sample concentration. Red dots: HCl + ACN; yellow dots: HCl + Hexane-EtOAc 1; light green dots: HCl + Hexane-EtOAc 4; dark green dots: HCl + MeOH; cyan dots: NaOH + ACN; blue dots: NaOH + Hexane-EtOAc 1; purple dots: NaOH + Hexane-EtOAc 4; Red violet dots: NaOH + MeOH. The vertical lines are the standard deviation of each extraction method. The metabolite concentrations were the semi-quantitative log values of identified metabolites.

**Figure 6 toxics-10-00688-f006:**
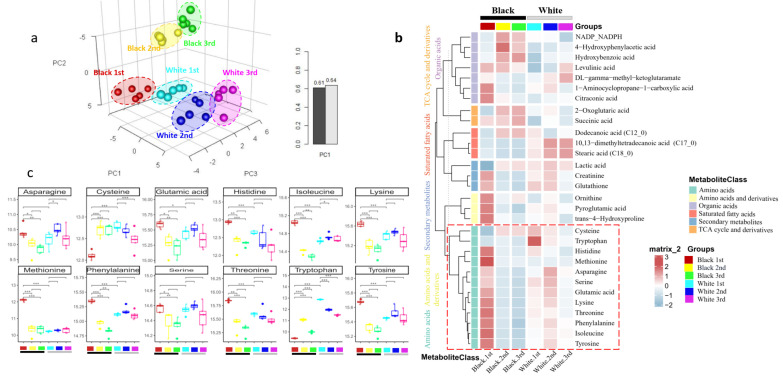
Significant metabolites in in three different segments of panda black and white fur samples. (**a**): Partial least squares-discriminant analysis (PLS-DA) of healthy pregnancy controls; first-black fur segments (Black 1st, red), second-black fur segments (Black 2nd, yellow), third-black fur segments (Black 3rd, green), first-white fur segments (White 1st, cyan), second-white fur segments (White 2nd, blue), third-white fur segments (White 3rd, fuchsia). Leave-one-out cross-validation for PLS-DA. (**b**): Heat map shows metabolites that differed significantly across the segments of black and white panda fur samples. Red colors represent higher metabolite concentrations, while blue colors indicate lower metabolite levels. The horizontal black bar indicates the black fur group, while the horizontal grey bar means the white fur group. The relative concentration of metabolite was scaled to have a mean of 0 and a standard deviation of 1 (z-score). Only metabolites with *p*-value less than 0.05 between segments are shown. (**c**): The box plot shows amino acids that differed significantly across the segments of black and white panda fur. *, **, *** mean different salience.

## Data Availability

The data that support the findings of this study are available from the corresponding author upon reasonable request.

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
