# Peer review of "Evaluating Different Extraction Approaches for GC-MS Based Metabolomics Analysis of the Giant Pandas’ Fur"

_toxics, 2022, doi:10.3390/toxics10110688_

Round 1

Reviewer 1 Report

Toxics

Review of Manuscript Number: toxics-1993026

TITLE: Evaluating Different Extraction Approaches for GC-MS Based Metabolomics Analysis of the Giant Pandas' Fur

General Comments

This manuscript addressed the metabolomics analysis of Giant Panda for the evaluation of endo- or/and exogenous toxic effects. The GC-MS carried out the assessment, the extraction protocol was optimized. The proposed protocol could be used to evaluate future metabolomic information from biological samples. The manuscript has an overall description and discussion, and I would like to suggest accepting this paper after the authors have addressed the following comments.

Major comments

(1)                   In metabolomics analysis, LC-MS(/MS) has been frequently used to determine metabolites. Generally, the derivatization of the chemical species, such as polar metabolites, does not need LC-MS measurement. Why did the authors select the GC-MS for the analysis in this study? Please insert the GC-MS measurement's reason or asset than LC-MS analysis in the Introduction Section.

(2)                   Similar to the analysis of drugs and metabolites in hair, it has been suggested that nails can be helpful as an alternative species to assess long-term exposure effects and other factors (e.g., Analytica Chimica Acta, volume 948, 2016, 40-47 and Molecules, volume 23(12), 2018, 3231.). Listing it in the Introduction Section as a control paper is expected to improve the quality of this paper.

(3)                   (Section 3.4.) Is the low drug or metabolite concentration in white hair due to a mechanism similar to humans? We recommend that a brief discussion be added to the Giant Panda's fur based on the findings of the process (a mechanism) of the uptake of chemical species into the hair.

Minor comments

(4)                   Page 4 Line 145

Please insert the gravity acceleration (x g) in the centrifuge process.

(5)                   Page 4, Line 151 - 163

There are minor errors. Please insert a space between numerical value and unit, and correct the s “-1” in Line 163.

(6)                   Page 10

Please re-check reference No. 23.

I hope that my comment is useful for the improvement of the article.

Author Response

Response to Reviewer 1 Comments

Major comments

  1.  In metabolomics analysis, LC-MS(/MS) has been frequently used to determine metabolites. Generally, the derivatization of the chemical species, such as polar metabolites, does not need LC-MS measurement. Why did the authors select the GC-MS for the analysis in this study? Please insert the GC-MS measurement's reason or asset than LC-MS analysis in the Introduction Section.

The authors agree with the reviewer’s suggestion. We have incorporated the underlying reason for choosing the GC-MS-based approach in the Introduction Section (lines 88-93):

“Furthermore, gas chromatography-mass spectrometry (GC-MS) based metabolomics is the most prevalent analytical platform to study human hair profile [7, 10, 17-19]. This preference is likely due to its high separation capability in analyzing polar metabolites, even though these polar compounds must be derivatized prior to analysis [20]. Electron impact (EI) ionization in GC-MS generates highly reproducible mass spectra independent of vendors [20], making public sharing and exchanging of GC-MS libraries available. Thus, The aim of the present work was to develop and validate an accurate sample preparation method for fur metabolism analysis to measure the exposome and metabolome of panda fur using gas chromatography-tandem mass spectrometry (GC/MS) based on methyl chloroformate (MCF) derivatization.”

  1. Similar to the analysis of drugs and metabolites in hair, it has been suggested that nails can be helpful as an alternative species to assess long-term exposure effects and other factors (e.g., Analytica Chimica Acta, volume 948, 2016, 40-47 and Molecules, volume 23(12), 2018, 3231.). Listing it in the Introduction Section as a control paper is expected to improve the quality of this paper.

We have addressed the reviewer’s comments in the introduction section (lines 72-74):

“Moreover, fur also contains functional metabolomes such as amino acids and lipids that could reflect endogenous metabolism [7]. Similar to the analysis of hair metabolome, it has been suggested that nails can be utilized as an alternative specimen to evaluate long-term environmental exposures [8, 9].”

References:

  1. Kuwayama, K., et al., Three-step drug extraction from a single sub-millimeter segment of hair and nail to determine the exact day of drug intake. Anal Chim Acta, 2016. 948: p. 40-47.
  2. Takahashi, F., et al., High-Frequency Heating Extraction Method for Sensitive Drug Analysis in Human Nails. Molecules, 2018. 23(12).
  3. (Section 3.4.) Is the low drug or metabolite concentration in white hair due to a mechanism similar to humans? We recommend that a brief discussion be added to the Giant Panda's fur based on the findings of the process (a mechanism) of the uptake of chemical species into the hair.

The authors thank the reviewer's comments. We have discussed why lower concentrations of metabolites were observed in white panda fur in section 3.4 Validation experiment:

“The level of amino acids was higher in black 1st segment, such as asparagine, glutamic acid, histidine, isoleucine, lysine, methionine, phenylalanine, serine, threonine, and tyrosine, while only cysteine and tryptophan were decreased (Fig 6c). Author et al demonstrated that human black hair contains brown-black eumelanin, a polyanionic indolequinone-based polymer, which facilitates hair matrix attraction with other metabolites or drugs through ionic interactions [29]. Thus, black fur may exhibit a higher affinity to retain metabolites than white fur.”

Minor comments

  1. Page 4 Line 145

Please insert the gravity acceleration (x g) in the centrifuge process.

The authors thank the reviewer’s suggestion. We have replaced all centrifugation units from rpm to gravity acceleration (x g) in Materials and Methods section 2.6 line 151: “Samples were centrifuged at 3220 x g for 15 min”.

  1. Page 4, Line 151 - 163

There are minor errors. Please insert a space between numerical value and unit, and correct the s “-1” in Line 163.

The authors apologize for the topographical errors. We have inserted a space between numerical value and unit as well as superscript “-1” in the Materials and Methods section in lines 155-160 and 169:

“In brief, 200 μL 1_M NaOH was added to resuspend, followed by 167 μL MeOH and 34 μL pyridine. 20_µL MCF was added to initiate derivatization by vigorous mixing for 30 s, and then another 20_µL MCF was added followed by another 30_s of mixing. 400_µL chloroform and 400_µL of 50_mM sodium bicarbonate were added and mixed for 10 s to separate the derivatized metabolites.”

“Scan speed was detected with 1.562 µ.s-1”.

  1. Page 10

Please re-check reference No. 23.

We have removed reference No. 23.

Reviewer 2 Report

Summary:  Giant Panda housed in Chinese zoos are exposed to air pollutants.  The goal was to identify metabolites present in Panda fur.  Black and white fur was extracted with 8 different solvent mixtures, derivatized, and subjected to GC-MS. The most efficient extraction solvent varied with the category of metabolite. Amino acids were extracted best with HCl-acetonitrile, while fatty acids were most efficiently extracted with NaOH-methanol.  Metabolites in black fur were somewhat different from metabolites in white fur. 

Minor comments:

1.     Table S2 and Figure 4a.  Typing error: cabamic acid should be carbamic acid.

2.     Figure 4a.  The words are difficult to read.  Please improve the resolution to make the letters in each word clear and readable.

3.     Figure 4.  The flower pot is a very nice way to summarize results.  However, the numbers and words in the flower pot are almost unreadable. 

4.     Figure 4.  The color of each method has a specific color in Figure 4a.  But these colors are not reproduced in the flower pot.  The flower pot uses a set of different colors for each extraction method.  It is recommended to use a consistent set of colors in Figures 4a and 4b.

5.     Figure 4 legend.  Typing error “the flower pot shows the number of identified numbers for each extraction method”.  The intended wording is “number of identified metabolites for each extraction method”.

6.     Figure 4 legend.  Please explain the basis for the metabolite lists.  For example, all 8 boxes are checked in the A list.  I interpret this to mean the metabolites in the A list include all metabolites found in all 8 extraction methods.  However, this assumption is inconsistent with lists B to L.  For example, the 3 metabolites in the L list are not in the A list.  Each list is unique.  How is a metabolite assigned to a list?

7.     Figure 4.  I do not understand what is graphed in Figure 4b.

8.     Figure 6b heat map.  Please write the words in a resolution that is readable. 

9.     Section 3.2 line 223.  typing error.  “UpSet plot (Fig. 4c)” should be UpSet plot (Fig. 4b)

10.  Section 3.4 line 277 missing word “between black and fur” should be “between black and white fur”

11.  Table S2 lists 18 amino acids identified in Panda fur.  Two amino acids, glutamine (Q) and arginine (R) were not found.  Please comment on why these 2 amino acids could not be detected by GC-MS. 

Author Response

Response to Reviewer 2 Comments

Minor comments:

  1. Table S2 and Figure 4a. Typing error: cabamic acid should be carbamic acid.

The authors apologize for the typographical errors. We have now corrected the spelling mistake in Table S2 and Figure 4a.

  1. Figure 4a. The words are difficult to read.  Please improve the resolution to make the letters in each word clear and readable.

We have improved Figure 4a to make each word readable, as displayed below:

  1. Figure 4. The flower pot is a very nice way to summarize results. However, the numbers and words in the flower pot are almost unreadable. 

We have enlarged the font size for the flower port in Figure 4. Please refer back to the improved imaging in question 2.

  1. Figure 4. The color of each method has a specific color in Figure 4a. But these colors are not reproduced in the flower pot. The flower pot uses a set of different colors for each extraction method. It is recommended to use a consistent set of colors in Figures 4a and 4b.

The authors thank the reviewer’s recommendation. We have re-illustrated the flower pot with a consistent set of colors in Figures 4a and 4b. Please refer back to the improved imaging in question 2.

  1. Figure 4 legend. Typing error “the flower pot shows the number of identified numbers for each extraction method”. The intended wording is “number of identified metabolites for each extraction method”.

We have now corrected the typographic errors in Figure 4 caption:

“b) The flower plot shows the number of identified metabolites for each extraction method.”

  1. Figure 4 legend. Please explain the basis for the metabolite lists. For example, all 8 boxes are checked in the A list. I interpret this to mean the metabolites in the A list include all metabolites found in all 8 extraction methods. However, this assumption is inconsistent with lists B to L. For example, the 3 metabolites in the L list are not in the A list.  Each list is unique. How is a metabolite assigned to a list?

The authors apologize for the confusion. We have added more explanation for the basis of the metabolite list in Figure 4 caption:

“a) Upset plot presents relationships between interactive sets of eight different extraction methods. The horizontal bar graph indicates the number of metabolites detected in each extraction method. The individual or connected black dots indicate the metabolite intersections that were either unique or shared between extraction methods. The vertical bar graph represents the number of shared metabolites. The metabolite list A-L display shared or unique metabolites between extraction methods, as shown by checked boxes (√). b) The flower plot shows the number of identified metabolites for each extraction method. Red : HCl+ACN; yellow: HCl+Hexane-EtOAc 1; light green: HCl+Hexane-EtOAc 4; dark green: HCl+MeOH; cyan: NaOH +ACN; blue: NaOH+Hexane-EtOAc 1; purple :NaOH+Hexane-EtOAc 4; and red violet: NaOH +MeOH.

  1. Figure 4. I do not understand what is graphed in Figure 4b.

For this question, please refer to our reply to question 6.

  1. Figure 6b heat map.  Please write the words in a resolution that is readable. 

We have improved the resolution of Figure 6b to make the words readable.

  1. Section 3.2 line 223.  typing error.  “UpSet plot (Fig. 4c)” should be UpSet plot (Fig. 4b)

We have now correctly labelled “UpSet plot (Fig. 4a)” in section 3.2 line 230.

  1. Section 3.4 line 277 missing word “between black and fur” should be “between black and white fur”

We have added the missing word in section 3.4 line 287: “between black and white fur”.

  1. Table S2 lists 18 amino acids identified in Panda fur.  Two amino acids, glutamine (Q) and arginine (R) were not found.  Please comment on why these 2 amino acids could not be detected by GC-MS.

We have added the comment in section 3.1 lines 195-196:  

“Nevertheless, glutamine and arginine levels in panda fur were below the detection limit of our method.”
